# Spatio-Temporal Bayesian Models for Malaria Risk Using Survey and Health Facility Routine Data in Rwanda

**DOI:** 10.3390/ijerph20054283

**Published:** 2023-02-28

**Authors:** Muhammed Semakula, François Niragire, Christel Faes

**Affiliations:** 1I-BioStat, Hasselt University, 3500 Hasselt, Belgium; 2Centre of Excellence in Data Science, Bio-Statistics, College of Business and Economics, University of Rwanda, Kigali 4285, Rwanda; 3Rwanda Biomedical Center, Kigali 7162, Rwanda; 4KIT Royal Tropical Institute of Amsterdam, 1092 AD Amsterdam, The Netherlands; 5Department of Applied Statistics, University of Rwanda, Kigali 4285, Rwanda

**Keywords:** malaria, survey data, routine data, spatio-temporal models

## Abstract

Introduction: Malaria is a life-threatening disease ocuring mainly in developing countries. Almost half of the world’s population was at risk of malaria in 2020. Children under five years age are among the population groups at considerably higher risk of contracting malaria and developing severe disease. Most countries use Demographic and Health Survey (DHS) data for health programs and evaluation. However, malaria elimination strategies require a real-time, locally-tailored response based on malaria risk estimates at the lowest administrative levels. In this paper, we propose a two-step modeling framework using survey and routine data to improve estimates of malaria risk incidence in small areas and enable quantifying malaria trends. Methods: To improve estimates, we suggest an alternative approach to modeling malaria relative risk by combining information from survey and routine data through Bayesian spatio-temporal models. We model malaria risk using two steps: (1) fitting a binomial model to the survey data, and (2) extracting fitted values and using them in the Poison model as nonlinear effects in the routine data. We modeled malaria relative risk among under-five-year old children in Rwanda. Results: The estimation of malaria prevalence among children who are under five years old using Rwanda demographic and health survey data for the years 2019–2020 alone showed a higher prevalence in the southwest, central, and northeast of Rwanda than the rest of the country. Combining with routine health facility data, we detected clusters that were undetected based on the survey data alone. The proposed approach enabled spatial and temporal trend effect estimation of relative risk in local/small areas in Rwanda. Conclusions: The findings of this analysis suggest that using DHS combined with routine health services data for active malaria surveillance may provide provide more precise estimates of the malaria burden, which can be used toward malaria elimination targets. We compared findings from geostatistical modeling of malaria prevalence among under-five-year old children using DHS 2019–2020 and findings from malaria relative risk spatio-temporal modeling using both DHS survey 2019–2020 and health facility routine data. The strength of routinely collected data at small scales and high-quality data from the survey contributed to a better understanding of the malaria relative risk at the subnational level in Rwanda.

## 1. Introduction

Malaria is a life-threatening disease ocuring mainly in developing countries. Almost half of the world’s population was at risk of malaria in 2020. Some population groups are at considerably higher risk of contracting malaria and developing severe disease: infants, children under five years of age, pregnant women, and patients with HIV/AIDS, as well as people with low immunity moving to areas with intense malaria transmission such as migrant workers, mobile populations, and travelers. The World Health Organization reported an increase of 6.2% in cases from 2019 (227 million cases) to 2020 (241 million cases). In the same period, the number of deaths due to malaria increased by 12.4% as compared to the previous year. African countries contribute about 95% of all malaria cases and 96% of deaths. Children under five are more vulnerable compared to the elderly, with 80% of all malaria deaths occurring in Africa [1].

Four countries in Africa count over half of malaria deaths globally: Nigeria (31.9%), the Democratic Republic of the Congo (DRC) (13.2%), the United Republic of Tanzania (4.1%), and Mozambique (3.8%) [1]. Those four countries count for 80% of all malaria deaths in Africa. East African countries such as Burundi, the DRC, Kenya, Rwanda, South Sudan, the United Republic of Tanzania, and Uganda are among the second most vulnerable countries [1]. The deaths and burden of malaria cases among children under five years is a leading public health concern. Malaria prevalence among children under five years in Burundi was 21.8% in 2012, and a recent Demographic and Health Survey (DHS) in 2017 showed an increase to 37.9%. For Rwanda, the last two DHS reported a decrease in malaria prevalence among children under five from 7.8% in 2015 to 2.7% in 2020 [2]. In 2015 malaria was the second cause of morbidity in Rwanda, representing 7.4% of outpatient consultations, and the sixth cause of mortality, representing 4.3% of total mortality. However, the Malaria Indicator Survey (MIS) conducted in 2017 showed a high prevalence of 11.8% in Rwanda. In Tanzania, the prevalence was 14.4% for DHS 2016 and slightly lower at 7.3% for MIS 2017. Uganda estimated a prevalence of 30.4% based on DHS of 2016, and MIS reported 18.2% in 2019 [2]. The prevalence at the subnational level is locally higher within countries based on DHS, as Figure 1 shows. In the Burundi’s northeastern regions bordering Rwanda and northeastern Uganda bordering South Sudan, malaria prevalence among under-five-year old children is above 60%.

East African community (EAC) member states (Burundi, Kenya, Rwanda, Tanzania, Uganda, South Sudan, and the DRC) have invested in malaria prevention and treatment strategies for decades and have implemented various prevention strategies. The strategies focused on distributing insecticide-treated mosquito nets (ITNs) and employing indoor residual spraying (IRS) within the households, and some countries facilitate citizens to access free or minimized primary healthcare services through community health insurance schemes. However, the malaria risk is still high, and countries must accelerate the implementation of life-saving strategies. The World Health Organization elaborated a global technical strategy for malaria elimination by 2030 that was built on three pillars: (1) Ensure universal access to malaria prevention, diagnosis, and treatment; (2) accelerate efforts towards elimination and attainment of malaria-free status; and (3) transform malaria surveillance into a core intervention.

Malaria surveillance is a core component of accelerating progress toward malaria elimination by 2030. All East African member states are malaria-endemic countries, and the level of malaria prevention and control measures depends on country-specific strategies. Figure 1 shows that areas of Burundi bordering Rwanda are high-risk malaria zones as well as areas bordering Uganda, South Sudan, and Kenya. Since malaria can easily cross borders, it is vital to establish joint effective and active surveillance that enables early detection of outbreaks at subnational levels and continuous assessment of interventions to guide changes for improved interventions.

All EAC member states use DHS findings to plan for malaria elimination including for localized regions within the country. However, the DHS does not provide information about time since it is a cross-sectional survey [3]. Indeed, DHS surveys are methodologically designed to provide estimates statistically powered at a specific domain, not necessarily at the lowest subnational levels (DHS, most of the time, is limited to the second subnational level). The locally-tailored response requires having real-time malarial estimates at the lowest administrative levels. In addition, DHS or Malaria Indicator Survey (MIS) surveys do not allow temporal analysis because DHS is often conducted every five years and MIS every two years. However, malaria patterns vary weekly or monthly, which implies immediate action to control malaria outbreaks. DHs/MIS are of high-quality standards and are representative through random selection and two-stage sampling approaches.

This paper combines information from Rwanda Demographic and Health Survey (RDHS) data from 2019–2020 and health facility routine data from 2019–2020. Individual malaria cases of each sector are not routinely recorded in Rwanda so the accurate number of malaria cases is unknown, even though a five-digit number is given in the Table 1, and instead, estimates done by each sector are kept in a health information system as the routine data, and solely those estimates have been used in other malaria analyses. Part of the DHS data was collected during the COVID-19 pandemic, and little is known about the disruption and how the exercise was affected. Similar to the health facility routine data, part of malaria cases were reported during the COVID-19 period. The COVID-19 pandemic resulted in patients increasingly seeking care in the community and posed challenges to maintaining the delivery of malaria services in Rwanda [4].

Rwanda developed a theory of change progressing from a lack of access to data and the absence of a culture of learning to a state of routine data utilization to identify and drive change in public health decision making [5]. Both DHS and health facility routine data include community-reported information. Figure 2 shows Rwanda’s primary data sources structure. The survey data are often considered the gold standard compared to routine data from health facilities. The routine data are used within health facilities for disease monitoring. Still, most countries use only data from surveys for public health decisions [6], despite the availability of a huge amount of routine data at a local level.

In this paper, we propose a method to account for both national surveys and routinely collected health facility data to estimate malaria relative risk incidence in small areas. Surveys are specifically designed to collect data on a specific topic and can be conducted with high levels of precision and accuracy. Updating routine data estimates with survey data can lead to more accurate and reliable estimates, especially for smaller geographic regions or subpopulations that may not be well represented in routine data. Updating routine data estimates with survey data can help to fill in gaps in the data and provide a more complete picture of the situation. Surveys can be designed to collect data on specific subpopulations or geographic regions that are not well represented in routine data. Updating routine data estimates with survey data can help to increase the coverage and representativeness of the data. Surveys can be designed to collect detailed information on a specific topic, allowing for more nuanced and in-depth understanding of the phenomenon being studied. Updating routine data estimates with survey data can help provide a more complete picture of the situation and support more informed decision-making.

This allows for both timely and local surveillance of malaria. A Bayesian spatio-temporal model is used for the analysis of the data.

## 2. Materials and Methods

### 2.1. Data Source

We extracted the routine health facility data from the Rwanda Health Information Management System (HMIS). These data include the number of malaria cases by health facility per month. The malaria cases are disaggregated by sex and age. More than 416 health facilities in Rwanda record malaria-positive cases in their respective catchment areas in the sector. This study focused on cases of malaria among children who are under five years old in each sector of Rwanda (416 sectors) for 2019 and 2020. The Rwanda Biomedical Centre granted permission to use the de-identified data for this research.

Rwanda Demographic Health Survey (RDHS) is a second data source. DHS is a nationally representative two-stage cluster sample designed to provide national, province, and district-level population and health indicators. Villages were the primary sampling unit (PSU), with 500 PSUs sampled, stratified by district. A total of 12,949 households were sampled. Our analysis focused on children under five who tested positive during the survey. The participants provided informed consent before participating in the survey. We were granted permission by the MEASURE DHS project (Authorization letter number 113068) to use these de-identified data for this analysis [7]. Table 1 summarizes the characteristics of the data used in this paper.

HHD stands for household details; the DHS collects household information, mainly demographic, education of parents, mosquito net use, etc.

### 2.2. Statistical Analysis

We suggest modeling malaria relative risk using surveys and routine data together to improve estimates. We modeled malaria risk in a two-step procedure: (1) fitting a binomial model using survey data, then extracting ${\hat{P}}_{x_{i}}$; (2) assigning mean ${\hat{P}}_{x_{i}}$ at a location (administrative level) in the Poison model as an effect.

#### 2.2.1. Binomial Model

For modeling malaria prevalence using DHS, we specified the model to predict malaria prevalence in Rwanda using the stochastic partial differential equation (SPDE) approach and integrated nested Laplace approximation with Bayesian inference.

The number of positive children under five years old $Y_{i}$ out of $N_{i}$ sampled people follows a Binomial distribution, with $P(x_{i})$ being the prevalence at location *i*
(i=1,2,3,⋯n)
$$Y_{i}|P\left(x_{i}\right)∼Binomial\left(N_{i},P\left(x_{i}\right)\right),$$
(1)$$logit\left(P\left(x_{i}\right)\right)=\beta_{0}+\beta_{1}Altitude_{i}+S\left(x_{i}\right).$$
where $\beta_{0}$ denotes the intercept, $\beta_{1}$ is the coefficient of altitude, and S() is a spatial random effect that follows a zero-mean Gaussian process with Matérn covariance function $\mathrm{Cov}\left(S\left(x_{i}\right),S\left(x_{j}\right)\right)=\frac{\sigma^{2}}{2^{\nu -1}\Gamma \left(\nu \right)}\left(\kappa |\right|x_{i}-x_{j}{\left||\right)}^{\nu }K_{\nu }\left(\kappa |\right|x_{i}-x_{j}||).$ In this formula, $K_{\nu }\left(\right)$ is the modified Bessel function of second kind and order ν>0 is the smoothness parameter, $\sigma^{2}$ denotes the variance, and κ>0 is related to the practical range $\rho =\sqrt{\frac{8\nu }{\kappa }}$, which is the distance at which the spatial correlation is close to 0.1.

We refer to this model as model (Equation 1) (M1).

#### 2.2.2. Modeling Routine Data Only

For modeling malaria relative risk using health facilities routine data only, we followed the approach proposed in the Bayesian spatio-temporal model for malaria risk in Rwanda [8].

Let $Y_{jgr}$ be a number of under-five-year old malaria cases, the period in months j=1,⋯,12, gender g=1,2, and sectors r=1,⋯416. The malaria cases data were counted and assumed to follow a Poisson distribution with the rate $n_{jgr}\lambda_{jgr}$, where $n_{jgr}$ is the associated population count. Thus, the linear predictor is given by the following equation:(2)$$\eta_{jgr}=\log\left(\lambda_{jgr}\right)=\theta_{g}+\varphi_{jg}+u_{r}+v_{r}$$
where $\theta_{g}$ is the gender-specific effect, and $\varphi_{jg}$ is a gender-period effect.

The last two terms are related to the spatial domain. Structured and unstructured spatial components were considered in this model; thus, u=1,⋯416 stands for spatial structured and v=1,⋯416 for unstructured components. This model is referred to as model (Equation 2) (M2).

The unstructured spatial random effects, $v={\left(v_{1},\cdots \cdots ,v_{416}\right)}^{T}$, are assumed to be independent and identical (iid) normally distributed with and unknown variance $\sigma_{v}$. Thus, the formulation can be written as iid $v_{r}$∼$N(0,\sigma_{v}^{2})$ [9].

The neighboring regions might have a similar malaria risk. Therefore we model $u={\left(u_{1},\cdots \cdots ,u_{416}\right)}^{T}$ using an intrinsic Gaussian Markov random field (GMRF). The Besag model includes a spatial random effect. The spatial random effects are assigned an intrinsic conditional autoregressive (ICAR) prior, which can be specified in a general form as $u_{i}|u_{j},i\neq j,\tau ∼N\left(\frac{1}{n_{i}}\sum_{i∼j}u_{j},\frac{1}{n_{i,\tau }}\right)$, where $n_{i}$ is the number of neighbors of node i, and *i*∼*j* indicates that the two nodes are neighbors. We defined the neighboring matrix as spatial proximity of the locations where malaria cases have been observed or are being predicted. The neighboring matrix was used to define a spatial weighting matrix, which is used to weight the contributions of neighboring locations to the prediction of malaria risk at a given location. For more details see [10,11].

For unstructured spatial effect v, we assigned a gamma distribution with a shape equal to 0.5 and rate equal to 0.00149, such as $\sigma_{V}^{2}$∼G(0.5,0.00149). For the other parameters, we used marginal standard deviations to assign a prior distribution to scaled precision parameters.

#### 2.2.3. Modeling Routine Data with Survey Data

To improve the estimates of malaria in children under five years old in Rwanda, we added information from RDHS to the analysis of routine data. We did this by adding the estimated prevalence based on DHS as a non-linear function in model M2. To do so, we needed an estimate of malaria prevalence $\hat{P}$ from model M1 at the sector level. This was obtained by averaging the posterior mean from model M1 at locations x∈r, i.e., $\hat{P}=\frac{1}{n(x\epsilon r)}\sum_{x\epsilon r}\hat{P}\left(x\right)$, and the model is given as:(3)$$\eta_{jgr}=\log\left(\lambda_{jgr}\right)=\theta_{g}+\varphi_{jg}+f\left({\hat{P}}_{r}\right)+u_{r}+v_{r}$$
where $f({\hat{P}}_{r})$ is modeled as an RW2 model after grouping ${\hat{P}}_{x_{i}}$ in 20 bins.

This model is referred to as model (Equation 3) (M3).

### 2.3. Bayesian Inference

The Bayesian hierarchical model in which prior distributions were assigned to all model parameters was applied. Independent diffuse priors were used for gender-specific intercepts, and independent smoothing priors for the gender effects. Then, a smoothing prior based on the second-order walk was assigned to each parameter. Recall that a second-order random walk penalized deviations from a linear trend. It was used to increase the precision [12]. The Deviance Information Criterion (DIC) proposed by Spiegelhalter [13] as a tool for model selection in a Bayesian context was used in this analysis. It is written as:(4)$$DIC=D(\bar{\theta })+2pD$$
where D() is the deviance and $\bar{\theta }$ is the posterior mean vector of unknown parameters of the model. The deviance measures how well the model fits the data, with larger values corresponding to a worse model. The parameter $P_{D}$ is the adequate number of parameters and measures the complexity of the model. The parameter is defined as the difference between the posterior mean of the deviance and the deviance evaluated at the posterior mean of the parameters. The rule of thumb for using DIC in model selection is roughly the same as for Akaike Information Criteria (AIC) and Bayesian Information Criteria (BIC); namely, a difference in DIC of more than ten rules out the model with higher DIC, while with a difference of less than five, there is no clear winner [14]. In order to verify whether the results obtained are not dependent on underlying prior specifications, sensitivity analysis was performed. Four different configuration priors were constructed to see whether estimates are not sensitive to priors. The conditional predictive ordinate for each observation was computed to check for the possible outlying observations in $y_{j}$ [15].

## 3. Results

### 3.1. Data Exploratory Analysis

The malaria incidence rate for children under five years old increased among both males and females in 2020 as compared to the incidence rate in 2019. The incidence rate increased from 50 to about 250. Figure 3 shows a consistent pattern of the positivity rate among males and females. The positivity rate was relatively constant in 2019. In 2020, the pattern of malaria transmission began to rise from September, reached its highest point in November. The implementation of indoor residual spraying (IRS) in the three districts of Rwanda with the highest risk of malaria, Kirehe, Ngoma, and Nyagatare, in September 2019 and January 2020, has been reported to have a significant impact on reducing the incidence of new malaria cases in a short amount of time [16,17].

For DHS 2019–2020, A total of 3665 children under five years old were selected in households surveyed and tested for malaria. Out of 3665 children tested, only 99 tested positive for malaria in Rwanda.

### 3.2. Main Findings

The main findings are presented in two parts. The first part provides malaria estimates from the binomial model, and the second provides estimates from joint Spatio-temporal models. We discussed the advantages of using survey data to improve health facility routine data estimates for decision-making in the era of the malaria elimination goal.

For the survey data, model M1 was used. Note that this is a spatial model, as DHS is a cross-sectional design. For the hospital routine data, models M2 and M3 were compared using DIC. M2 had a DIC of 831,071, while M3 had a DIC of 822,394. Therefore, model M3 is a more preferred model. The model used in this study is spatio-temporal, as monthly routine data were collected and analyzed over both space and time. In order to address the limitations of routine data, the study also incorporated less frequently obtained Demographic and Health Survey (DHS) data. The spatio-temporal M3 model, which incorporated the DHS data, was found to be superior to the M2 model that did not use the DHS data.

The M3 model provides a more comprehensive understanding of malaria trends, as it can capture both short-term and long-term changes over space and time. Therefore, the spatio-temporal M3 model is a valuable tool for monitoring and predicting malaria incidence in a given area.

Table 2 shows the parameter estimates of these final models.

The posterior summaries, including the posterior mean (estimate) and its standard deviation (sd), lower limit (LL), and upper limit (UL), are presented.

Figure 4 shows spatial variation of malaria prevalence among under-five-year old children in Rwanda.

The estimation of malaria prevalence among those under five years old using Rwanda demographic and health survey data 2019–2020 showed a higher prevalence in the southwest, central, and northeast of Rwanda compared to the rest of the country. The upper limit (UL) prevalence map shows a similar pattern to the prevalence estimated though the intensity is different. The UL shows the worst scenario. The hot spot areas were identified using the exceeding probability of prevalence above 10%.

Figure 5 shows spatial variation of malaria relative risk incidence among under-five-year old children in Rwanda using DHS and routine health clinical data combined. Figure 5 provides additional clusters as compared to the results in Figure 4; the last one did not show the southern areas of Rwanda among the high-risk malaria zones. Using M3, we identified new malaria clusters in the southern area of Rwanda for the years 2019 and 2020. Those clusters had a more significant RR than 35 (RR >35), with an increase from 2019 to 2020 in those specific clusters. The relative risk (RR) quantifies whether the area *i* has a higher or lower occurrence of cases than the expected from reference rate. The estimated RR quantifies the heterogeneity of the risk and highlights unusual patterns of risks. The upper and lower limits are provided through the 95% credible interval.

Figure 5 shows Rwanda’s southern, southeastern, southwestern, and central regions as high-risk malaria areas. The estimates are very localized at the sector level with greater precision than the DHS data. The M3 provides the advantage of detecting new clusters not identified using only DHS survey data. We identified more than nine sectors (Mamba, Gishugi, Muganza, Mukingo, Mugombwa, Nyanza, Nyanza, Kansi, Nora, Nyanza) with RR >40, and two sectors (Gishubi, Muganza) with an RR ≥ 50 in southern areas of Rwanda that were not detected using DHS survey 2019–2020 data.

This modeling approach enables the use of all available information and minimizes the loss of information. For this case, we can also evaluate malaria’s relative risk over months. Malaria risk is very localized and requires more localized interventions. We compared malaria risk in January, June, and December for 2019 and 2020. We observed that overall malaria incidence increased in 2020, but sectors at risk were fewer (30 sectors) than those at risk in 2019 (52 sectors).

We assessed the posterior temporal trend effect for malaria relative risk ($\exp(\phi_{t}+\gamma_{t}$) for condition autoregressive (CAR) for 2019 and 2020. Figure 6a,b shows CAR models with a 95% credible interval.

The general trend of malaria risk reveals a rise in malaria risk from September to December 2019, and then a decrease from April to July 2020. However, there was an increase in cases in August, reaching a peak in November 2020.

The 95% credible trend effects intervals were narrow for 2019 and 2020. We observed no difference in risk between males and females. This approach to modeling malaria relative risk indicates a bottleneck analysis of establishing a Malaria trend surveillance system that provides evidence that public health experts can use to elaborate an area-specific strategy. It is crucial to use M3, which combines survey and health facility routine data to enhance or improve detailed and localized estimates at sector levels.

## 4. Discussion

Our results present a novel approach to modeling health facility data and survey data to improve estimates of malaria relative risk in small areas. Most developing countries use DHS data as a primary data source to estimate key health indicators such as malaria prevalence. In the last decade, there have been emerging findings recommending the use of routinely reported clinical data for program planning and evaluating impact. However, most of those studies reported data quality concerns as a limitation, mainly data completeness, timeliness, accuracy, consistency, and poor utilization of Health Management Information System (HMIS) tools [18]. The introduction of web-based information systems for health facility data and the implementation of universal health policy improved the completeness and accuracy of data at the local level and population-based statistics based on those data. The most recent studies have reported improved data quality and accuracy of HMIS in some countries [8,19,20]. Sub-Saharan Africa (SSA) countries have adopted the HMIS that includes data quality control checks to ensure high quality. Rwanda, as one of the East African countries, has been using HMIS since 2011. Generally, data from health facilities are of high quality due to consistent data quality checks and feedback between national and decentralized levels [8]. Ideally, health programs are built on evidence-based practice through successfully integrating data into health policy. Decision making throughout the health system is an ongoing challenge.

Low- and middle-income countries use DHS data as an alternative source for planning and evaluating impact. DHSs are cross-sectional studies conducted every two or five years. The DHS data are of high quality and enable comparison of studies using DHS data to ascertain commonalities or unusual patterns and to compare countries or regions. DHS surveys provide population-based coverage data of key health service indicators. These indicators can be disaggregated, and differentials can be assessed according to geographic, biodemographic, and socioeconomic characteristics. However, DHS survey data have some limitations. Desegregation in small geographic areas is desirable in malaria elimination stages since the unit of implementation of health programs or intervention is dynamic downwards. DHS surveys are generally not designed to yield estimates of health indicators at the sector level since this is too costly. The frequency of availability of new data is between two and five years. Lastly, DHS data does not allow trend analysis [3,21]. DHS surveys remain an important source of health data in developing countries at the national and international levels.

Different solutions were proposed to improve estimates from health facility routine data, from more straightforward approaches such as data triangulation, developing a robust algebra estimator by adjusting for testing practices, care-seeking behavior, and reporting completeness to complex models that account for uncertainties [8,22,23,24]. Both health facility routine and survey data are important sources of health data; therefore, using both data sources is crucial.

In this paper, we proposed a novel approach of extracting fitted values from a binomial model using survey data and using them as random effect covariates in the poison model to estimate the relative risk of malaria at sector administrative levels and evaluate trends over time. The M3 model incorporating DHS estimates in modeling malaria relative risk incidence using routine case data yielded a substantial variation within and between sectors’ relative risk. The overall trend effect of malaria relative risk varied by month. We observed two sensitive malaria risk periods in Rwanda, namely April–June and September–December. The model allows to evaluate malaria relative risk per month, in contrast to cross-sectional surveys. The model takes advantage of cross-sectional survey data to improve area-specific and time-specific estimates from routine data.

Though the time effect component is vital to understand malaria epidemiology, area-specific risks can inform the national malaria control program to conduct effective and efficient strategies for reducing malaria incidence at the subnational level. Our findings show malaria relative risks per sector (administration level 3) in Rwanda.

The areas detected through combining DHS survey and routine data (Mamba, Gishugi, Muganza, Mukingo, Mugombwa, Nyanza, Nyanza, Kansi, Nora, Nyanza, Gishubi, Muganza, and Mukingo in the southern areas of Rwanda) neighbor high-risk malaria areas in Burundi, as Figure 1 shows. Not detecting a high-risk area might lead to classifying southern areas of Rwanda among areas at low risk of malaria during the planning process and a failure to develop area-specific interventions. The findings show an overall increase in malaria in 2020, but the distribution of cases was more localized in a few sectors as compared to the previous year, 2019. DHS survey data are often used in Rwanda for planning and evaluating the impact of interventions conducted to prevent malaria [25]. The malaria incidence from a model using surveys and routine health facility data allows a comparison of incidence between sectors and an evaluation of trends over time. The two-step modeling approach allowed us to quantify the trends in changes in relative risk incidence in every small area over time.

The modeling approaches presented in this paper enable estimates at small administrative areas that only DHS data cannot provide because DHS surveys are designed and empowered to provide estimates at the domain level. Most of the time, the domain levels in the survey are second-higher administrative entities, such as districts in Rwanda. A survey such as DHS is costly; sampling in small areas might not be practical due to time and cost related. DHS data collection does not include all months of the year; it usually targets high and low-malaria-risk months in a year. Though the approach of targeted sample collection on space and time is important, the left-out areas and months during sample collection for seasonal diseases might be a limitation in capturing the true picture of the disease in small areas. The Rwanda DHS 2019–2020 data collection process occurred in November and December 2019 and January, February, March, June, and July 2020. The posterior trend effect showed that malaria risk rose in April, May, September, and October. The peak of malaria risk was in June and December. The two-step modeling approach can provide a complete picture of malaria in space and over time using both survey and routine data. The strength of both sources of data overcomes the limitations of both sources. The health facility routine data have a limitation in quality and completeness. Routine data might not include those who did not seek health services in health facilities and in the community. Malaria is primarily managed by Community Health Workers (CHWs) at the village level who are trained to provide essential health services in their communities in Rwanda [26]. DHS is a household survey, and there is an equal chance of sampling both people who seek health services and those who do not. Therefore, taking advantage of routine and DHS survey data through statistical modeling contributes to a better understanding of the malaria burden. The proposed modeling approaches enable monthly trend effect analysis and small-area disease mapping.

The World Health Organization (WHO) issued a new final phase of malaria elimination and related appropriate strategies in June 2022. Three categories of possible interventions were identified: (1) “mass” strategies applied to the entire population of a delimited geographical area such as a district, sector, cell, or village; (2) “targeted” strategies applied to people at increased risk of infection compared to the general population; and (3) “reactive” strategies implemented in response to individual cases [27]. All three strategies would benefit from active surveillance with real-time data. The WHO and groups of experts recommend the establishment of a malaria real-time data surveillance system to eliminate malaria [8,27]. It is crucial to empower routine data with DHS estimates to establish an active malaria surveillance data system with real-time data.

The modeling approach presented in this paper contributes to the utilization of all data sources to gain improved estimates at small areas and quantify the trend effects.

## 5. Conclusions

In summary, we recommend using both the DHS survey and routine data in a two-step modeling framework for spatio-temporal models as one of the tools for active malaria surveillance to identify high-risk areas, monitoring malaria trends, prediction of new cases and evaluation of malaria incidence at subnational levels. This approach enables the use of cross-sectional survey data and health facility routine data in modeling relative malaria risks, as well as generating malaria risk maps that provide information on the spatio-temporal trend effects of malaria at subnational levels. The proposed approach is not limited to malaria cases and can also be applied to other infectious disease monitoring and evaluation at small areas.

We discussed the advantage of using multiple data sources in a two-step modeling framework. We modeled malaria relative risk among under-five-year old children in Rwanda. We compared findings from geostatistical modeling of malaria prevalence among under-five-year old children using demographic and health survey 2019–2020 (M1) data and modeling malaria relative risk using both DHS survey 2019–2020 and health facility routine data. The model provided the estimates at sector levels and enabled analysis of malaria relative risk trend effects. The findings show that the model that uses DHS survey and routine data (M3) identified more high risk areas in the southern province of Rwanda that were not detected using only survey data.

Both survey and routine data are used to establish efficient permanent monitoring of the variation of malaria risks in areas and evaluate interventions towards the final phase of malaria elimination. The strength of routinely collected data at small scales and high-quality data from the survey should contribute to a better understanding of the malaria burden and support evidence-based decision making.

## Figures and Tables

**Figure 1 ijerph-20-04283-f001:**
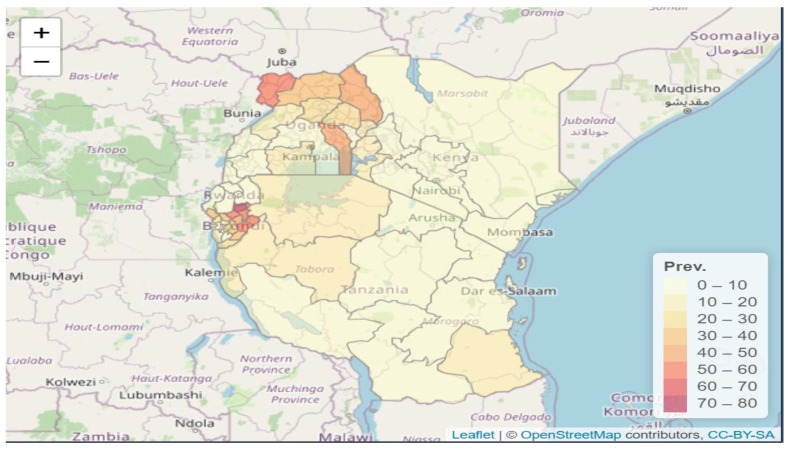
Percentage of children of age 6–59 months that tested positive for malaria in East Africa (Burundi, Kenya, Rwanda, Tanzania, and Uganda at subnational. Source: DHS data).

**Figure 2 ijerph-20-04283-f002:**
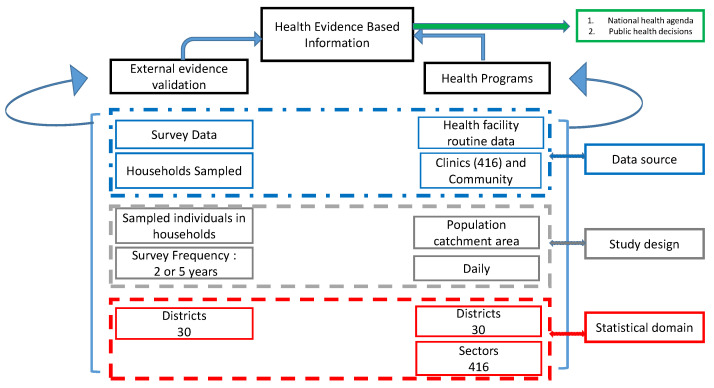
Data sources to inform evidence-based decision in Rwanda.

**Figure 3 ijerph-20-04283-f003:**
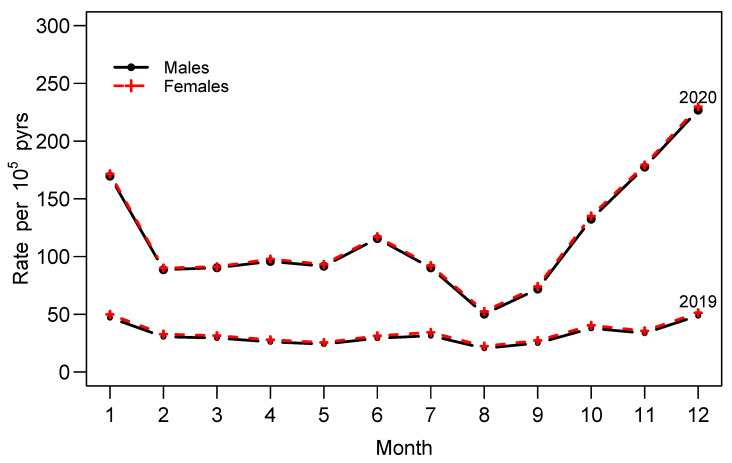
Incidence rate of malaria among children under five years old per 100,000 population. Health facility routine data for year 2019 and 2020.

**Figure 4 ijerph-20-04283-f004:**
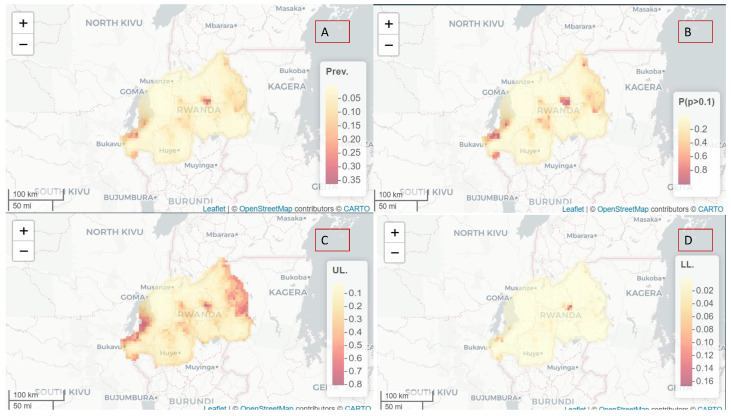
Spatial variation of malaria prevalence among children under five years old in Rwanda, RR posterior means of spatial effects (**A**) with 95% UL upper limit (**C**). Lower limit (**D**). Panel (**B**) shows exceeding probability of prevalence above 10% (**B**). Source: DHS 2019-2020.

**Figure 5 ijerph-20-04283-f005:**
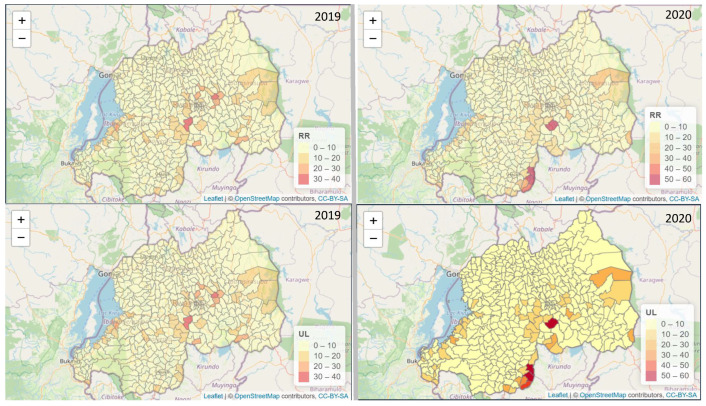
Spatial variation of malaria relative risk incidence among children under five years old in Rwanda (RR), posterior means of spatial effects with 95% upper limit (UL). Source: health facility routine data from 2019 and 2020.

**Figure 6 ijerph-20-04283-f006:**
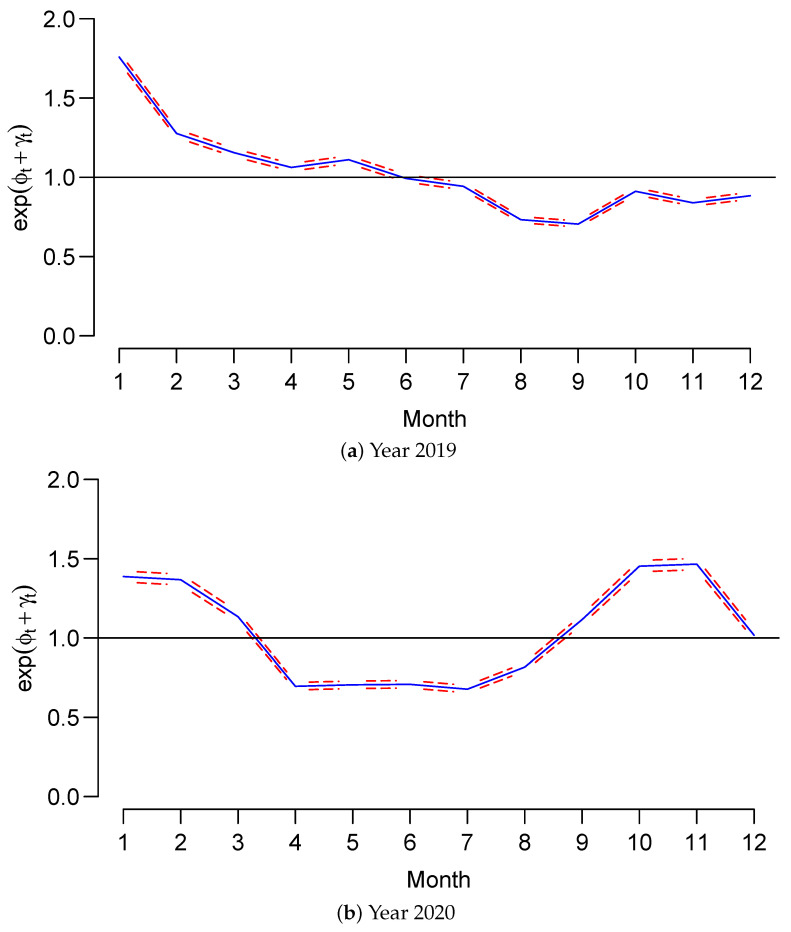
Posterior temporal trend effect for malaria relative risk: $\exp(\phi_{t}+\gamma_{t})$ with 95% credible interval, year 2019.

**Table 1 ijerph-20-04283-t001:** Datasets used for children under five years old.

Type	Cases	Sample	Domain	Freq	Covariates
Routine	39,936		Sector (416)	Daily/Month	Gender, location
DHS	99	3665	District (30)	Cross-sectional	Gender, HHD, location
Shapefile			Sector (416)		
Population size			Sector (416)		

**Table 2 ijerph-20-04283-t002:** Malaria risk parameter estimates for M1 and M3.

Models	M1			M3		
**Parameters**	**Est (sd)**	**LL**	**UL**	**Est (sd)**	**LL**	**UL**
Fixed Effect						
$\mu_{0}$	−2.584 (1.069)	−4.796	−0.588			
$\mu_{Male}$				−5.892 (0.047)	−5.984	−5.800
$\mu_{Female}$				−5.865 (0.047)	−5.957	−5.773
$\beta_{1Altitude}$	−0.002 (0.001)	−0.003	0.001			
Random Effect						
$u_{r}$	−4.62 (0.268)	−5.14	−4.62	0.584 (0.124)	0.380	0.866
$v_{r}$	2.69 (0.186)	2.33	3.06	1.242 (0.192)	0.905	1.658
$\varphi_{j}$				15.59 (4.72)	8.025	26.420
$varf({\hat{P}}_{r})$				3.072 (0.975)	1.538	5.336

## Data Availability

All data used in this paper can be accessed at https://github.com/semakulam (accessed on 5 May 2022).

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
