# Peer review of "Spatio-Temporal Bayesian Models for Malaria Risk Using Survey and Health Facility Routine Data in Rwanda"

_ijerph, 2023, doi:10.3390/ijerph20054283_

Round 1

Reviewer 1 Report (Previous Reviewer 3)

The new version of the manuscript "Spatio-Temporal Bayesian Models for Malaria risk using survey and health facility routine data in Rwanda" has improved. Most of my concerns expressed for the first version were solved, but a few minor corrections are required. Also, the English language and text style need a professional revision to better address several issues that compromise the understanding across the text.

Author Response

Reviewer 2 Report (Previous Reviewer 2)

0) l40 blue part. Under WHAT years old?

1A) Please address the ignored instruction "l38-56. Describe the source of each fact presented here (currently none provided)". Particularly: l39-41. If the source of each fact described here is ref [2], insert "[2]" between "Mozambique (3.8%)" and ". Those", mention "demographic and health survey" and define DHS here. Otherwise, give a correct source of data.

1B) l46-47. What is the source of the 2012 data? If it is DHS and the source of data in l39-41 is not DHS, mention "demographic and health survey" and define DHS here. Otherwise, give a correct source of data.

1C) l47. If the source of the 2017 data collected by "a recent demographic and health survey" is ref [2] but the sources of data mentioned in l39-41 and l46-47 (2012 data) are not, insert "[2]" after "increase to 37.9%". If the source of the 2017 data is not [2], give a correct source of data instead.

1D) l47. Define "DHS" here, not in l48.

1E) l48. Change "Demographic and Health Surveys (DHS)" to "DHS", because the definition must have been already done in either l39-41 of l47.

1F) l48-50. If the source of the data collected by "a recent demographic and health survey in 2017" is ref [2] but ref [2] has not yet been cited before, insert "[2]" after "2020". Otherwise, give a correct source of data instead.

1G) I accessed the Spatial Data Repository (spatialdata.dhsprogram.com) (ref [2]). Malaria-related data for Rwanda seem not be available except there for those of "Use of mosquito nets by children" collected in DHS surveys done in 1992, 2000, 2005, 2008, 2010, 2013 (MIS), 2015, and 2019 (no malaria case number data in any year I could find there). I think it would be better to explain more practical details how to get malaria prevalence data and data collected by 2016 and 2017 surveys from there, probably in Materials and Method section. If there is another proper source, give it instead of [2].

2) l40. Give definition of "DRC" at the first appearance of "Democratic Republic of the Congo" and use exclusively the abbreviation thereafter. (eg. change l43)

3) l88. MIS has been defined already in l52. Use the abbreviation only.

4) l89-90. Is "DHS is often conducted every five years and MIS every two years" correct? (l301-2?)

5) l142. Delete "in individual"

6) l148, 161, 187. "where" start from lower case w. Do not indent (see l190 which is correct).

7) l158. Change "count" to "counted"

8) l163. Change "u = u1, ....416" to "u = 1, ....416" and "v = v1, ....416" to "v = 1, ....416"

9) l167 and 176. I guess the position of 2 over σ is better to be slightly more right compared to v as is now because it probably means σv x σv

10) l167. Is it OK that the suffix v is a capital V?

11) l176 "r.,i.e" to "r, i.e."

12) l187. "xi" to "xi"(i suffix to x)?

13) between l189 and l190. "linear trend. it" to "linear trend. It" ("I" capital).

14) l190. "D(.)" to "D( )" (no "dot" in ( ) ). I don't understand why the authors ignored suggestion of this correction given at the previous round of review.

15) Sentences in Result, Discussion, and Conclusions sections.  In general, results of a research are supposed to be described with verbs in the past tense as they were observed/obtained in the research. By contrast, general facts which are not dependent on particular research should be described with verbs in the present tense. Check each sentence in these sections carefully if it is a description of an observation or a general fact, and correct if necessary.

16) l214. Change "the figure" to "Figure"

17) l215. Delete "In 2020 there ... to 2019" as it has been already mentioned in l206-213. Instead, write something telling that rate was relatively constant in 2019 here.

18) l216-8.  Figure 3 does not show anything "Malaria increases ... decline in August." Rewrite these to correctly express the fact in 2020 and put that in front of Figure 3, not after the figure like now.

20) l218-20 blue sentences.  Perhaps better to move these two sentences to the front of Section 3.1.

21) Figure 5 legend. Give explanation of RR and UL - perhaps, insert "(RR)" after "Rwanda", delete ":", change "Posterior" to "posterior", replace "CI" with "(UL)", and change "," to "."???

21) l249. Isn't "The estimates are very localized at the sector level with precision." obvious as the number of data is much, much higher in sector level data? Does this tells anything about precision?

22) l249-50. "The M3 ...DHS survey data." Same as 21) above.

23A) l259. "but sectors at risk were fewer than those at risk in 2019". Give a particular number of each year for direct comparison. That would help also discussion in l358-9.

23B) Do the boundaries between sectors next to next act physical barriers more significant than simple distance to hinder movement of both humans and mosquitoes across the sectors? Is the prevalence of malaria uniform within a sector always?

23C) If the boundaries are easily crossable and the distribution of malaria cases in the sector is not uniform, what is the point to consider distribution of malaria at the sector level?

24) l265. Figures 6?

25) Figure 6. (1) The risk was very high at the beginning of year not only in 2020 but also in 2019 when the malaria case number was not so high (Figure 3). Why? (2) Is the gap in risk between December 2019 and January 2020 real or just artificial due to data being in different analysis years?

26A) Figure 6. (1) Put legend below the figure. (2) There are some interferences which disturb sight of some parts of this figure. Check each figure is complete or not before submit it. 

26B) Figure 6. There are no explanation of φt, ϒt, and exp(φt, ϒt).

27A) l326-7.  "We observed two sensitive malaria risk periods in Rwanda, April-June and September-December." Which data support this? How about high risk in January seen in both figures 3 and 6?

27B) Is it OK to ignore differences between different years? Is it sufficient to see only two year data to generalise the trend in a year?

28) Ref 28 in reference list looks odd, though it seems not be cited anywhere in the text.

29) Format of references in the reference list is not uniform.

30) This research confirmed that there are time (month, year)-dependent variations in malaria risk and suggested that it is probably useless to statistically estimate monthly trend for prediction of real risk in a different year (n is only 2, though). The area-specific and time-specific estimates can be obtained from regularly collected numerous routine data covering all subsections. Even after reading this revised manuscript, I don't understand the necessity to consider cross sectional data, which are taken only occasionally (once in 2-5 years?), to IMPROVE area-specific and time-specific estimates.

Author Response

Reviewer 3 Report (Previous Reviewer 4)

Comment to Authors

Title: Spatio-Temporal Bayesian Models for Malaria risk using survey and health facility routine data in Rwanda

Main Comment

Part of the strategies employed in malaria elimination includes being able to predict the disease burden especially risk to the finest detail demographically which help put in proper policies and management strategies. In this study, Semakula et al proposed a two-step modeling framework using survey and routine data to improve estimates of malaria risk incidence in small areas and enable quantifying malaria trend effects. This work was done in Rwanda in Africa which is a malaria endemic region. This was done by Bayesian spatio-temporal modeling malaria relative risk through the combination of data from the Rwanda Demographic and Health Survey data and health facility routine data from 2019-2020.

This is a much-improved version of the manuscript. The authors have put in a good effort in the analysis and model. I believe this is a good paper and will strongly encourage the author to take time and read over the manuscript again to make all necessary corrections.

The conclusions derived from the study are valid since they were comparing just before the pandemic as was as during the pandemic. This make the paper an important one to understand the dynamics of the SAR-COV-2 and COVID-19 burden on the eventual malaria outcomes modelled. The paper is recommended for publication subject to minor corrections.

Minor Comments

Abstract

Well done

Materials and Methods

The study area and selection of study population are well defined and fits the study. The procedure for collecting the health survey data are clearly stated. The formulas used for the modeling are also clearly stated with their underlying limitations.

Results

The results presented are well done.

Discussion

The discussion presented is scientifically sound and derived from the observation made in the study.

Conclusion

The conclusions derived from the study are valid and are derived from the work done.

Round 2

Reviewer 2 Report (Previous Reviewer 2)

Issues

1) l48.  Correct "deaths" to "prevalence"

2)  Thanks to the authors for giving explanation about "routine data" they handled. Until I saw that, I had misunderstood that in Rwanda, all malaria cases identified in each sector are routinely reported and collected to make a set of precise data which is referred as "routine data". However, now I understand that the "routine data" they refer in this paper were only estimates, not the accurate counts at all. Then, the authors can be right - inclusion of survey data can improve the estimate, even when the apparent number of data is way far less than those in the "routine data".

So, I would suggest the authors should:

2a)  Change the title of the paper to:

Spatio-Temporal Bayesian Models for Malaria Risk Using the Survey and the Health Facility Routine Data in Rwanda

2b)  In Introduction (possibly between "2019-2020." and "Part of" in l93), clearly state that:  

      2b-1) individual malaria cases of each sector are not routinely recorded in Rwanda so the accurate number of malaria cases is unknown, even a 5-digit number is given in the Table 1, and

      2b-2) instead, estimates done by each sector or the statistics office are kept in the national statistics as the Routine Data, and solely those estimates have been used in other malaria analysis.

2c)  Insert all (1-4) in the response in the cover letter given to my previous comment 30 to the Introduction section (around l107, not around l132) to justify incorporation of survey data for this analysis. Make it clear that health facility routine data of this country are not the actual counts but just estimates.

3a) l207. ", and then declined from December to March"???  This "March" is March 2021, isn't it? Nothing showing this is given in Figure 3. Delete this phrase.

Or, if the authors want to state about the decline observed in early 2020, they should describe it just after "In 2020," in l205. That decline had nothing to do with December 2020 data but must have had something to do with December 2019 data.  I am curious why the rate suddenly jumped to the value which was more than twice between December 2019 and January 2020.  If there is any theory to explain the jump, the author should provide it here.

3b) l207-209.  "This cycle repeated itself starting in mid-April, with another peak in transmission occurring in June, before finally declining again in August. The trend is similar in 2019 but the magnitudes are different."??? Same as above- nothing showing these is given in Figure 3. Delete these two sentences.

4) l221. "Note that this is spatio-temporal model, as routine data is collected every month". Because, in this study, the same DHS data is applied to compensate the weakness of the Routine Data, the M3 model was better than the M2 model which is free from the less-frequently-obtained DHS data. I suspect DHS data are not constant in long term. Is it OK to simply say that the M3 model is a spatio-temporal model which can explain trend of malaria for not only a short period of time (limited to within a year or 2) but also a longer term?

5) l221.  Change "the preferred model" to "a more preferred model than M2"

6) Figure 6.  Please discuss why there was a sudden, big difference in risk between December 2019 and January 2020. This is important because the authors state a continuous trend through the gap.

7) l257.  I suspect that the name of a month should appear between "from" and "2019" in this line. Fill it.

8)  The authors should incorporate the response to my previous comment 23B, to Materials and Methods. Also state what "specified threshold" which was set for this analysis.

Other things

9) Ref 2.  Add "https://" to make the URL complete.

10) l46.  Define "DHS" here, not in l47 as in current manuscript; use the abbreviation "DHS" instead of "Demographic and Health Surveys (DHS)" in current l47 .

11) l166.  Make "besag" to "Besag"

12) l182.  I do not understand why the authors answered "Thank you, addressed" without actually addressing my previous comment 11.

13) (No line number in this PCF; 5 line below l186)  I do not understand why the authors answered "Thank you, addressed" without actually addressing my previous comment 13.

14) l184.  I do not understand why the authors answered "Thank you, addressed" without actually addressing my previous comment 12.

15) l206.  Remove "typically"

16) Figure 5.  I do not understand why the authors answered "Thank you, replaced as suggested" without actually addressing my previous comment 21.

17 (just truly a comment: no need to answer) Why does the analysis year should be 12 months (J=1,2,..,12) and start from January and end in December in each year? I understand that data of months in the middle of year were useful to compare between 2019 and 2020, but for comparison between two consecutive months across the change of year, why do you have to do two separate analyses in each of which only part of the term of comparison is considered? Isn't it possible to consider all 24 months altogether or start analysis from middle of year, to discuss the trends across the change of year?

Author Response

This manuscript is a resubmission of an earlier submission. The following is a list of the peer review reports and author responses from that submission.

Round 1

Reviewer 1 Report

The work presented proposes a two-step mathematical based on survey and routine data for estimating the risk of malaria in endemic areas.  The authors used data from East Africa.  The approach is promising.  It would be important for the authors to comment on the application of their model in West Africa.  Would  it be applicable? 

Major comments:  The authors should carefully review the manuscript and include the figure, references and table numbers. Citation section is not properly placed. 

Reviewer 2 Report

This manuscript should be turned down as it is not yet ready to be peer reviewed. This is my first time to be asked to review such a preliminary draft. None of the reference papers are correctly cited in the text. In addition, figures' and tables' numbers missing, odd English, data missing, etc... Absolutely awful.

Nevertheless, I read this draft to the end anyway. I am not sure why DHS survey data should be included for rough estimation when sufficient number of precise routine data are available. The methodology written in this paper may not have sufficient value and may not contributes to malaria epidemiology significantly. To clear these suspects, the authors should clearly explain that the M2 model is not good enough and that the M3 model is definitely required for obtaining useful estimates.

The authors should demonstrate evidence that some important features could not identified with M2 but they are clearly identified with M3. It is totally insufficient to only briefly mention that M3 seems to be more preferable than M2 (l210). 

Otherwise, I suspect this methodology paper would have little value to be published in this journal, even when it has properly cited references/properly numbered figures and tables.

l9. alternative to what?

l38-56.  Describe the source of each fact presented here (currently none provided).

l38.  What does "Nigeria (31.9%)" mean?

l42.  1) Is the DRC one of East Africa countries?

2) If your "East Africa countries" is the East African Community (the EAC), define it and list the member state here, not in l42 or further below.

l45.  What does 21.8 mean?

l46.  Which is right, "Demographic" or "demographic?

l46-56.  Some of the values of prevalence shown here are much larger than those of the four more affected countries (l38-40). Is it because these countries have much less population compared to the four countries?

Figure 1.  Provide the data source.

l58.  1) Is your definition of EAC correct? Isn't it the East African Community (the EAC)?

2) If so, list up the member countries here, not in l78-79.

l71.  "All East African member states" is better to be rewritten to "All EAC member states".

l79.  DHS has been already defined in l46. Do not define more than once.

l85.  What are "DHS/MIS surveys"? Define.

l87.  What is DHs? What is "DHs/MIS"?

l92-94.  Rewrite the sentence "Rwanda developed...drive change [?]" to make sense.

l96.  "Figure ?? shows Rwanda’s primary data sources structure." If this mentions about Figure 2, the figure does not show the structure of data but illustrate the data collection system and the data collected. Rewrite the sentence.

Figure 2.  This seems to be redundant as there is Table 1 which is more informative and much less misleading than this. You can remove this, or keep it after doing improvement especially considering follows:

1) Letters in green box are too small. Make them as large as those in other boxes.

2) Data source.

2a) (left) Are "Survey Data" and "Households Sampled" different sets of data? If not, merge the boxes.

2b) (right) Are "Health facility routine data" and "Clinics (416) and Community" different sets of data? If not, merge the boxes.

3) Statistical domain.

3a) (both) Does "Districts 30" mean "30 districts"? (Same for Sectors)

3b) (right) Are District data and sector data independent? If not, merge the boxes.

3c) What is the domain of Community data mentioned in Data source?

4) To which data is each study design applied how?

5) What do the "[" and "]" at the sides indicate? What does each curved line with arrowhead mean?

6) What is Health Evidence Based Information? Isn't it Evidence-Based Health Information?

7) This figure doesn't tell anything about informing decision, and data source structure is only part of what it shows. Change the title to not misleading one.

Formula (4)  1) D in "2pD" is better to be shown pD, where D is a suffix of p so it should be in SMALL CAPITAL and slightly offset below compared to p.

2) Probably better to include a new Formula (5) which explains the definition of pD written in l183-185.

l180.  What is "D(.)"? Is it "D( )"?

l182.  Not "parameter PD" but "parameter pD". See above.

l186.  AIC? BIC? Do you definitely need these here? If so, define each (Akaike..., Bayesian...). Otherwise, delete.

l201-203.  For the 2 years data shown in the Figure 3, descriptions in these 2 sentences did not apply at all. How did you get these 2 sentences from your data?

l210.  Here it is written that M3 seems to be better than M2.

1) I wonder if M2 isn't good enough. What is the problem to use M2 but not M3?

2) If this just claims that M3 is better than M2, is there no alternative model which is better than M3? If no such one exists, clearly state "M3 is the best model". If not, why do you compare M2 and M3 only?

Table 2.  Why don't you include M2 in the table for comparison?

l228-229.  Are these clusters unidentifiable with the M2 model? If they can be sufficiently identified even with M2, what is the point to include DHS survey data for this analysis?

l230. Define RR (relative risk? risk ratio?) in the text. There is no explanation from what RR is calculated out how in Materials and Methods. Please provide it.

l246  You have defined ICAR before in l161 but not yet done for CAR in this text. Define it here.

l248-250.  These two sentences do not make sense. Rewrite.

l251.  "The 95% credible trend effects intervals were narrow for 2019 and 2020." Which data tells so?

l251-253. "The pattern of...cases, rain, and temperature." Where are the data that support these?

l251-260.  This is probably inappropriate being present in Results. Better to move to Discussion or Conclusions.

l268.  Define HMIS here, not in l273.

l272.  Why do you want to define the abbreviation SSA here? It is used only once here in this text. Better not use the abbreviation.

l274-275.  Use "EAC" instead of "East African"

l280.  Use only the abbreviation DHS without its write out

l281  "alternative" to what?

Reviewer 3 Report

 "Spatio-Temporal Bayesian Models for Malaria risk using survey and health facility routine data in Rwanda" presents a relevant topic as it proposes a model to predict malaria outbreaks in Rwanda. It has the potential to be refined and integrated with official surveys to help health authorities to establish more efficient strategies to combat malaria, as it is a severe disease that mostly affects African countries.

The manuscript raised a few questions. I will point out some of them - the attached file has more detailed comments.

1 - The references across the text are missing. Where should the number of the corresponding reference be there is a [?], which prevents reaching the exact reference and check information. Also, when a figure is mentioned across the text, the respective number is missing which affects comprehension;

2 - Introduction: the authors should add a brief explanation of the etiologic agent of malaria, its life cycle, and how it is transmitted. If the data is available, it would be beneficial to add the most prevalent Plasmodium species in the researched area. Also, this would contextualize the information provided in lines 307-322.

3 - Results: 

Lines 196-197 mention November to July, but there is no reference to August, September, and October. 

However, reading lines 200-203 I assume the survey followed the period of one year and, although is not indicated in figure 3, one can assume that "Period" refers to the months whose data were analyzed during one year. 

But if only November and December were evaluated in 2019, and, I suppose, January to October were evaluated in 2020, how there are monthly points if each year was not entirely analyzed?

Figure 6 presents a posterior monthly trend effect and the data presented is comprehensible, but here I had doubts.

I apologize if I might have misunderstood - I would like to clarify this aspect. Also, I suggest changing the numbers on the X-axis to the name of the corresponding month.

Lines 248-250: This sentence is confusing. The increase in malaria risk was from 2019 to December 2020?

Or there was an increase from December/2019 to April/2020, followed by a decrease in the relative risk that remained until July/2020?

4 - Discussion: 

* There is one sentence that suggests a relation between climate aspects with malaria cases, but does not "prove" it, as follows: 

Lines 252-254: "(...) malaria monthly trends is similar to the trend of precipitation and temperature. There is a relationship between malaria cases, rain, and temperature. However, a deep investigation of malaria cases, temperature, and rain relationship in Rwanda as a focus still needs to be performed."

Malaria monthly trends are shown, but the precipitation and temperature monthly trends do not seem to have been evaluated in the manuscript. 

This study does not seem to have investigated a correlation between malaria cases and climate aspects in Rwanda - I don't think this affirmation (lines 307-308) is adequate to this context, especially because the results don't mention the tendency os rainfall/mm of precipitation through the months.

Would this be an assumption because the months in which there is an increase in malaria parameters coincide with the ones with rainfall peaks? Because it does not seem it was a parameter evaluated in this research, but an affirmation made due to the results obtained.

If its the case, I would suggest reformulating the sentence as the relationship between climate aspects (rain and temperature) and malaria cases/peaks was not a focus in this manuscript.

* Lines 281-282: "DHSs are cross-sectional studies conducted every two or five years."

What factors influence the periodicity of the studies? I believe it would be interesting giving a brief explanation of how those studies are conducted and what influences the periodicity.

*Are there studies that proposed a model for malaria in other countries? Or for another infectious disease? I think the discussion misses this aspect and more literature should be discussed along with the results found.

Reviewer 4 Report

Comment to Authors

Title: Spatio-Temporal Bayesian Models for Malaria risk using survey and health facility routine data in Rwanda

Main Comment

Part of the strategies employed in malaria elimination includes being able to predict the disease burden especially risk to the finest detail demographically which help put in proper policies and management strategies. In this study, Semakula et al are proposing a two-step modeling framework using survey and routine data to improve estimates of malaria risk incidence in small areas and enable quantifying malaria trend effects. This was done by Bayesian spatio-temporal modeling malaria relative risk through the combination of data from the Rwanda Demographic and Health Survey data from 2019-2020 and health facility routine data from 2019-2020.

The text question marks all over that need to be re-edited especially for the references, figures and table names in the main text of the paper. This made it difficult to read and follow. Nevertheless the authors have put in a good effort in the analysis and model. I believe this is a good paper and will strongly encourage the author to make all necessary corrections.

The conclusions derived from the study are NOT totally valid. Once of the biggest drawbacks in the study in the lack of information on SAR-COV-2 and COVID-19 burden on the eventual malaria outcomes modelled. I would have been great is the author have modeled 2018 to 2021 which will be interesting to see a better trend irrespective of the COVID-19 burden. The reduction in malaria cases observed could be due to factors relating to the effect of the pandemic. In light of the above, I find it difficult as to why these factors were not considered or avoid totally by selecting an earlier date other than 2020 or a pandemic period.

I strongly recommend the work for corrections.

 Minor Comments

Abstract

The conclusions needs to be modified

Materials and Methods

The study area and selection of study population are well defined and fits the study. The procedure for collecting the health survey data are clearly stated. The formulas used for the modeling are also clearly stated with their underlying limitations. Missing is their information on the potential influence of the pandemic on malaria outcomes.

Results

The results presented OK.

Discussion

The discussion presented is NOT so scientifically sound although they are derived from the observation made in the study. The discussion should have included extensively the effect of COVID-19 in the eventual dataset used in the model. This in turn highly make these models a bit unscientifically valid since we can not be sure how the model would have turned out if these pandemic parameters were considered.

Conclusion

The conclusions derived from the study are NOT totally valid. These are sound and derived from the work done but they do require major revision.

Reviewer 5 Report

Greater emphasis given in this work concerns the more or less well used mathematical models over the years, even if I don't have the mathematical competence to judge the models, but I am sure that mathematicians do not always include all the factors upstream of the vector infestation, and they have not been able to attribute, for example, on the component of the effectiveness of the contrrol measures in lowering or increasing the vectors, therefore, lowering and raising the risk of malaria. If the importance of the models were true, as mathematicians say, we would have already had more convincing indications from their  results , and we would have prevented the emergence of cases of malaria. But I am convinced that this work is set up well and will have to demonstrate its effectiveness, yes or no. The use of this model  should be repeated to verify the usefulness of the model. The high profiles of the authors presages that we are dealing with a scientific work. We need to see over time if what they say has had an effect and rewrite another next work, once this study has been used. Good work and I look forward to see your next work done by use of this outcome. The quality of data used is still an issue , should it be taken in account .